# PET imaging-guided chemogenetic silencing reveals a critical role of primate rostromedial caudate in reward evaluation

Yuji Nagai[1], Erika Kikuchi[1], Walter Lerchner[2], Ken-ichi Inoue[3], Bin Ji[1], Mark A.G. Eldridge[2], Hiroyuki Kaneko[1], Yasuyuki Kimura[1], Arata Oh-Nishi[1], Yukiko Hori[1], Yoko Kato[1], Toshiyuki Hirabayashi[1], Atsushi Fujimoto[1], Katsushi Kumata[4], Ming-Rong Zhang[4], Ichio Aoki[5], Tetsuya Suhara[1], Makoto Higuchi[1], Masahiko Takada[3], Barry J. Richmond[2] & Takafumi Minamimoto[1]

The rostromedial caudate (rmCD) of primates is thought to contribute to reward value processing, but a causal relationship has not been established. Here we use an inhibitory DREADD (Designer Receptor Exclusively Activated by Designer Drug) to repeatedly and non-invasively inactivate rmCD of macaque monkeys. We inject an adeno-associated viral vector expressing the inhibitory DREADD, hM4Di, into the rmCD bilaterally. To visualize DREADD expression *in vivo*, we develop a non-invasive imaging method using positron emission tomography (PET). PET imaging provides information critical for successful chemogenetic silencing during experiments, in this case the location and level of hM4Di expression, and the relationship between agonist dose and hM4Di receptor occupancy. Here we demonstrate that inactivating bilateral rmCD through activation of hM4Di produces a significant and reproducible loss of sensitivity to reward value in monkeys. Thus, the rmCD is involved in making normal judgments about the value of reward.

[1] Department of Functional Brain Imaging, National Institute of Radiological Sciences, National Institutes for Quantum and Radiological Science and Technology, 4-9-1 Anagawa, Inage-ku, Chiba 263-8555, Japan. [2] Laboratory of Neuropsychology, National Institute of Mental Health, National Institutes of Health, Bethesda, Maryland 20892, USA. [3] Systems Neuroscience Section, Primate Research Institute, Kyoto University, Inuyama, Aichi 484-8506, Japan. [4] Department of Radiopharmaceuticals Development, National Institute of Radiological Sciences, National Institutes for Quantum and Radiological Science and Technology, 4-9-1 Anagawa, Inage-ku, Chiba 263-8555, Japan. [5] Department of Molecular Imaging and Theranostics, National Institute of Radiological Sciences, National Institutes for Quantum and Radiological Science and Technology, 4-9-1 Anagawa, Inage-ku, Chiba 263-8555, Japan. Correspondence and requests for materials should be addressed to T.M. (email: minamimoto.takafumi@qst.go.jp).

The rostromedial caudate (rmCD) is a subregion of the striatum that receives strong projections from frontal limbic areas, especially the orbitofrontal cortex (OFC)[1]. Recording studies in this region have revealed neuronal response signalling information concerning the expected reward size. The same population of neurons exhibits relatively weak selectivity to movements/actions[2,3]. These anatomical and physiological characteristics support the hypothesis that the rmCD is part of the brain's reward circuit[4]. However, a causative role of the rmCD in reward evaluation and decision-making has not been established in primates. Unlike sensory and motor representations that are strongly lateralized, representations of reward and related functions seem to have relatively little lateralization, and thus function can remain unperturbed when a critical brain region is damaged unilaterally[5]. Therefore, any inactivation approach should target the rmCD bilaterally.

Chemogenetic techniques, such as DREADDs (Designer Receptors Exclusively Activated by Designer Drugs), offer a means to repeatedly and non-invasively inactivate multiple brain sites simultaneously. When a modified human M4 muscarinic acetylcholine receptor (hM4Di) is expressed in a target neuronal population, activity of such neurons is temporarily suppressed after systemic delivery of a biologically inert inducer compound, for instance, clozapine-N-oxide (CNO)[6]. To achieve the intended chemogenetic silencing, the inhibitory DREADD, hM4Di, needs to be accurately localized to a target structure and expressed at a sufficiently high level to silence host neurons. In addition, the inducer compound must be available at a concentration high enough to bind and activate the DREADD. Previously, it was only possible to determine the extent of receptor expression through post-mortem histological analysis, and to determine ligand-binding characteristics ex vivo.

Here we report a non-invasive method for imaging DREADD expression in vivo using a PET ligand, [11]C-labelled clozapine ([11]CLZ)[7]. Macaque monkeys received injections of a DREADD-expressing viral vector into the rmCD, were scanned with [11]CLZ-PET to visualize the sites and level of DREADD expression in vivo, and then also with a blocking protocol to measure DREADD receptor occupancy by CNO (Fig. 1a,b). The resulting data can be applied to determine an appropriate dose of CNO for subsequent behavioural experiments with chemogenetic silencing. We show that CNO activation of the hM4Di receptor expressed bilaterally in the rmCD successfully produces a clear and reproducible decrease in sensitivity to reward value in monkeys.

## Results

### In vivo PET visualization of hM4Di expression in rmCD.
Three monkeys (#171, #184, #190) were trained to perform a 'reward-size' task (Fig. 1c). Each trial in this task began when the monkey touched a lever, which was followed by the appearance of a visual cue that signalled the size of the upcoming reward (1, 2, 4 or 8 drops; Fig. 1d). To obtain a liquid reward, the monkeys had to release the lever when a visual target changed colour from red to green. Here, as in previous studies, the monkeys' error rates (proportion of trials with incorrect responses; releasing the lever too early or too late) were related to the value of the upcoming reward[8].

The monkeys received bilateral injections of adeno-associated virus serotype 2 (AAV2), containing a construct with a cytomegalovirus (CMV) promoter that expresses the hM4Di receptor without protein tags (AAV-hM4Di) into the rmCD of both hemispheres (Fig. 1a; one pair of injections in #171 and #190; two pairs of injections in #184; see below regarding #184). The combination of AAV2 and CMV results in neuron-specific expression in monkeys[9]. We verified that activation of the hM4Di receptor with CNO caused neuronal silencing in vitro (Supplementary Fig. 1).

To localize and monitor the hM4Di expression in vivo, we applied [11]CLZ-PET. DREADD-PET visualization was validated with two other monkeys expressing hM4Di in one side of the striatum (monkeys #157 and #127; Supplementary Fig. 2). We confirmed that the striatal sites visualized by PET (ligand uptake >120% of baseline) were coextensive with those showing the hM4Di expression histochemically (Supplementary Fig. 2). The [11]CLZ-PET scans in monkey #171 showed increased [11]CLZ binding in the bilateral rmCD at day 45 post AAV-hM4Di, at which point the expression seemed to have reached a plateau (Fig. 2a,b).

### Chemogenetic inactivation of rmCD impairs reward estimation.
To determine the extent to which hM4Di receptor is occupied by CNO as a function of dose, we performed [11]CLZ-PET scans 10 min after i.v. bolus CNO doses of 1, 3 and 10 mg kg$^{-1}$ in monkey #157. Striatal uptake at the hM4Di-vector injection site diminished as the CNO dose increased, whereas uptake was unchanged on the contralateral control side (Fig. 2c). The relation between occupancy of hM4Di and CNO dose was approximated by a Hill function (Fig. 2d). We chose a CNO dose of 3 mg kg$^{-1}$, because it yields ~50–60% receptor occupancy of hM4Di, and is not converted to a significant amount of CLZ (Supplementary Fig. 3).

We examined behavioural effects of chemogenetic silencing of rmCD in monkey #171. We confirmed that the CNO treatment (3 mg kg$^{-1}$, i.v.) produced comparable receptor occupancy at rmCD (Fig. 2d,e). The CNO treatment increased the overall error rates compared with non-treatment or vehicle controls (two-way

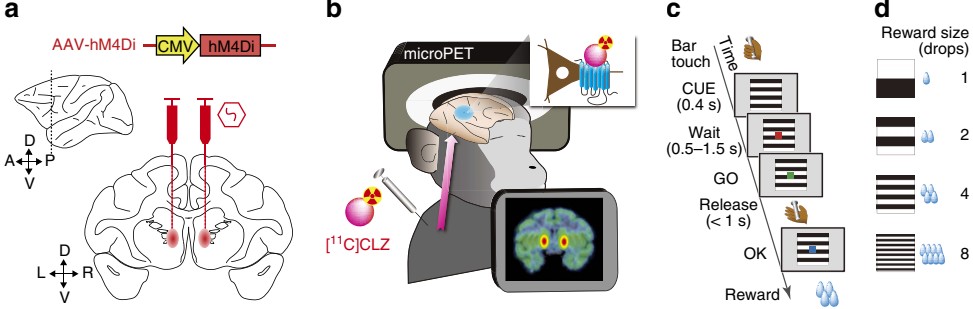

**Figure 1 | Experimental design.** (**a**) Three monkeys received AAV-hM4Di vector injections into the rmCD. (**b**) [11]CLZ-PET was performed to monitor DREADD expression in vivo and to measure occupancy of DREADD with CNO. (**c**) While monkeys performed reward-size task, CNO was administered intravenously to induce chemogenetic silencing in the rmCD. (**d**) Cue-reward-size pairing in reward-size task.

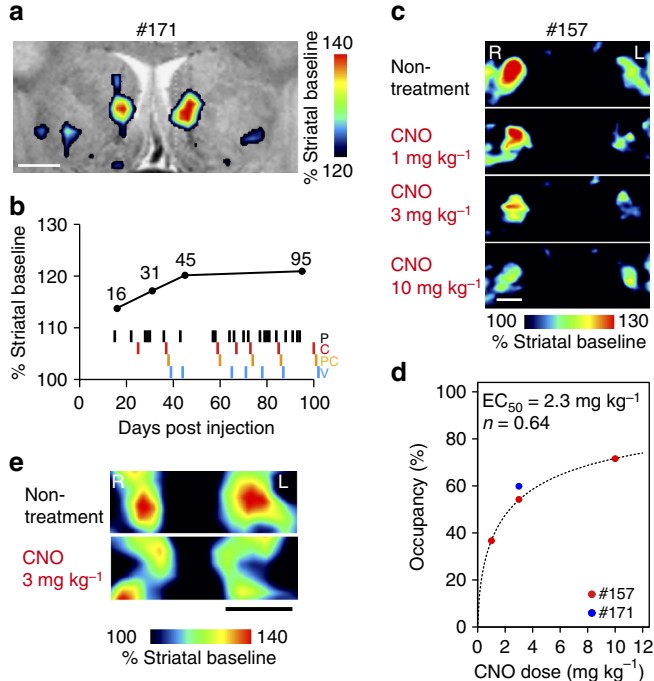

**Figure 2 | PET monitoring hM4Di expression in bilateral rmCD.**
(**a**) Coronal [$^{11}$C]CLZ-PET image taken 45 days after hM4Di vector injection (only for area over 120% striatal baseline), overlaid on MR image for monkey #171. (**b**) Time course of [$^{11}$C]CLZ uptake at the rmCD after vector injection for monkey #171. Coloured ticks indicate behavioural testing condition on the corresponding day: P, C, PC and V indicate post vector, CNO, post CNO and vehicle, respectively. (**c**) Coronal [$^{11}$C]CLZ-PET images at the bilateral putamen level in monkey #157. PET images at non-treatment and after the administration of 1, 3 and 10 mg kg$^{-1}$ doses of CNO (i.v.) are shown from top to bottom. (**d**) Occupancy of hM4Di receptor is plotted as a function of CNO dose. Red and blue represent data obtained from monkeys #157 and #171, respectively. Dotted curve is the best-fit Hill function to the data from #157. ED$_{50}$ and $n$ indicate the CNO dose achieving 50% occupancy and Hill coefficient, respectively. $R^2 > 0.99$. (**e**) [$^{11}$C]CLZ-PET images at bilateral rmCD level in monkey #171 at non-treatment (upper) and after administration of 3 mg kg$^{-1}$ doses of CNO (bottom). Scale bar, 5 mm.

analysis of variance (ANOVA), main effect of treatment, $F_{4,66} = 6.5$, $P = 1.6 \times 10^{-4}$; *post hoc* TukeyHSD, $P = 2.7 \times 10^{-11}$, CNO versus Post vector; $P = 3.4 \times 10^{-6}$, CNO versus Vehicle; Fig. 3a). The increase in error rates was observed throughout the testing session (that is, 15–115 min after CNO injection; three-way ANOVA, main effect of treatment, $F_{1,9} = 10.7$, $P = 0.0095$) in parallel with the increase in error over time (interaction with time, $F_{4,216} = 9.2$, $P = 0.073$; main effect of time, $F_{4,216} = 15.5$, $P = 3.4 \times 10^{-11}$) (Fig. 3d). On the day following CNO injection, the error rates had returned to baseline levels (*post hoc* TukeyHSD, $P = 0.20$; Post CNO versus Post vector; Fig. 3a), showing that the functional effect of CNO treatment lasted <24 h. There was no difference in the baseline error rates—i.e., in the absence of CNO—between pre- and post-vector injection (*post hoc* TukeyHSD, $P = 0.92$; Fig. 3a), suggesting that the behavioural effect of hM4Di expression alone was negligible.

In the other two monkeys (#190 and #184) the [$^{11}$C]CLZ-PET scans also showed increased [$^{11}$C]CLZ binding in the bilateral rmCD (Fig. 4a,b, day 29 after second injection in #184 and day 31 after injection in #190). The CNO treatment increased the overall error rates in these monkeys (main effect of treatment, $P < 0.01$, Fig. 4c,d).

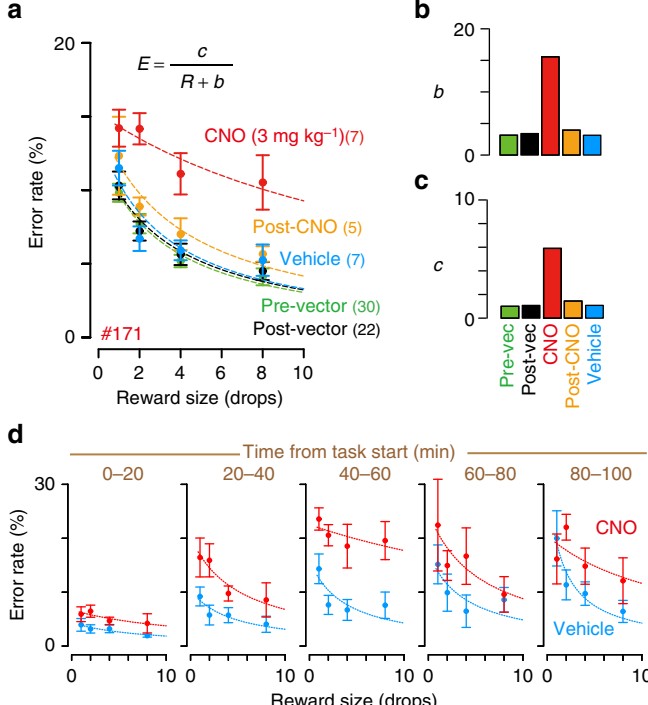

**Figure 3 | Behavioural effect of PET-monitored chemogenetic silencing.**
(**a**) Performance of reward-size task of monkey #171. Error rate (mean ± s.e.m.) as a function of reward size is plotted. Performance of pre-viral vector injection (Pre-vector, green), >15 days after viral injection (post vector, black), CNO treatment (3 mg kg$^{-1}$, i.v., red), on the day after CNO treatment (post CNO, orange), and treatment with vehicle without CNO (blue). Dotted curves represent best fit of the inverse model, $E = \frac{c}{R+b}$, where $E$ and $R$ are error rate and reward size, and $b$ and $c$ are free parameters shown in **b,c** that quantify the shift of inverse relation and the impact of reward size on error rates, respectively. $R^2 > 0.75$. (**d**) The time course of CNO induced behavioural effects in monkey #171. Error rate was plotted as a function of reward size for each 20-min period for CNO (red) and vehicle treatment (blue).

For all three monkeys, the reaction times in correct trials were not affected by CNO administration (two-way ANOVA, main effect of treatment, $F_{1,134} = 0.12$, $P = 0.72$). The monkeys generally reached satiety and stopped initiating new trials within the 100 min testing period. The total reward earned was slightly, not significantly, less in the CNO treatment sessions (ANOVA, main effect of treatment, $F_{1,32} = 3.1$, $P = 0.09$). These results suggest that the increase in overall error rates observed after chemogenetic silencing was not due to a general reduction in attention or drive.

To quantify the effect of rmCD silencing on reward evaluation, we analysed the relationship between the error rates and reward size. The error rates ($E$) in this task have an inverse relation with reward size ($R$)[8]. Here we used a modified version of the inverse function, $E = \frac{c}{R+b}$, where $c$ is a free-fitting parameter that quantifies reward requirement for a given performance and $b$ quantifies a shift to a flatter portion of the curve. This equation reportedly explains alterations of reward evaluation in monkeys with OFC ablations[10]. This equation also described the error rates in the current study ($R^2 = 0.91 \pm 0.09$, mean ± s.d., for example, Fig. 3a, dotted curves). Compared with vehicle treatment, CNO treatment increased parameter $b$ (Fig. 3b, CNO), reflecting a loss of sensitivity to reward contrast. CNO treatment also increased parameter $c$ (Fig. 3c, CNO), indicating that more reward is needed for correct performance (that is, increasing overall error

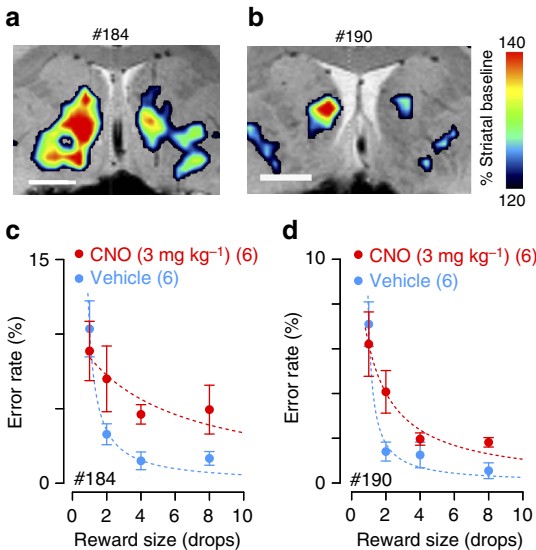

**Figure 4 | Chemogenetic silencing of rmCD produced loss of reward sensitivity.** (**a,b**) *In vivo* hM4Di expression in monkeys #184 and #190, respectively. Coronal image of [$^{11}$C]CLZ-PET overlaid on structural MR image, taken at day 31 and day 29 after vector injection, respectively. Scale bar, 5 mm. PET images for area over 120% striatal baseline are shown. (**c,d**) Effect of chemogenetic silencing of rmCD on performance of reward-size task. Error rate (mean ± s.e.m.) as a function of reward size is plotted for CNO treatment (red) and vehicle control sessions (blue). Dotted curve is the best fit of inverse function ($R^2 > 0.80$).

rates). Thus, chemogenetic inactivation of the rmCD leads to decreases in the sensitivity to reward.

In the two control monkeys without AAV-hM4Di vector injections, CNO treatment alone had no effect on performance (main effect of treatment, $P > 0.05$; Supplementary Fig. 4a). In another two monkeys that received AAV-HA-hM4Di vector injections into the rmCD bilaterally, the [$^{11}$C]CLZ-PET scans failed to detect increased uptake at the injection sites (Supplementary Fig. 4b, top; day 29 and 38 after vector injections, respectively). Subsequent CNO administration did not affect error rates in these monkeys (main effect of treatment, $P > 0.05$; Supplementary Fig. 4b, bottom). One of the three monkeys with a behavioural effect due to the DREADD inactivation initially failed to show any effect of CNO treatment on performance (#184; main effect of treatment, $F_{1,4} = 0.006$, $P = 0.94$; Supplementary Fig. 4c, bottom). A [$^{11}$C]CLZ-PET scan exhibited increased uptake in one hemisphere that was located dorsal to the intended target (Supplementary Fig. 4c, top; day 31 after the 1st AAV-hM4Di injection). This led us to give a second set of injections, after which a PET scan showed increased uptake that covered the intended rmCD target (Fig. 4a), and we then obtained the behavioural results described above (Fig. 4c).

After several CNO treatments, [$^{11}$C]CLZ-PET still showed increased uptake in the bilateral rmCD (#171, 121%, day 95, Fig. 2b; #184, 125%, day 83 after second injection), suggesting that repetitive CNO treatment in our study did not produce significant on-going tachyphylaxis of hM4Di-DREADD receptors. DREADD activation once a week is insufficient to rule out tachyphylaxis on a shorter timescale.

**Comparison with pharmacological inactivation.** To compare the behavioural effects of chemogenetic inactivation with those of pharmacological inactivation, we injected the GABA$_A$ receptor agonist muscimol into the rmCD in two other monkeys (#181 and #182; Fig. 5a). X-ray computed tomography (CT)

images visualizing injection cannulae confirmed that the sites of muscimol injection (Fig. 5b–d) corresponded to those of hM4Di expression (cf. Fig. 2a). For both monkeys, pharmacological inactivation of the bilateral rmCD increased the overall error rates (main effect of treatment, $P < 0.01$; Fig. 5c,d, bottom) and diminished the sensitivity of reward, as indicated by increases in parameters $b$ and $c$ (Fig. 5e,f, muscimol). Unilateral inactivation of the rmCD produced a trend toward increased overall error rates (main effect of treatment, $P = 0.065$; Supplementary Fig. 4d),

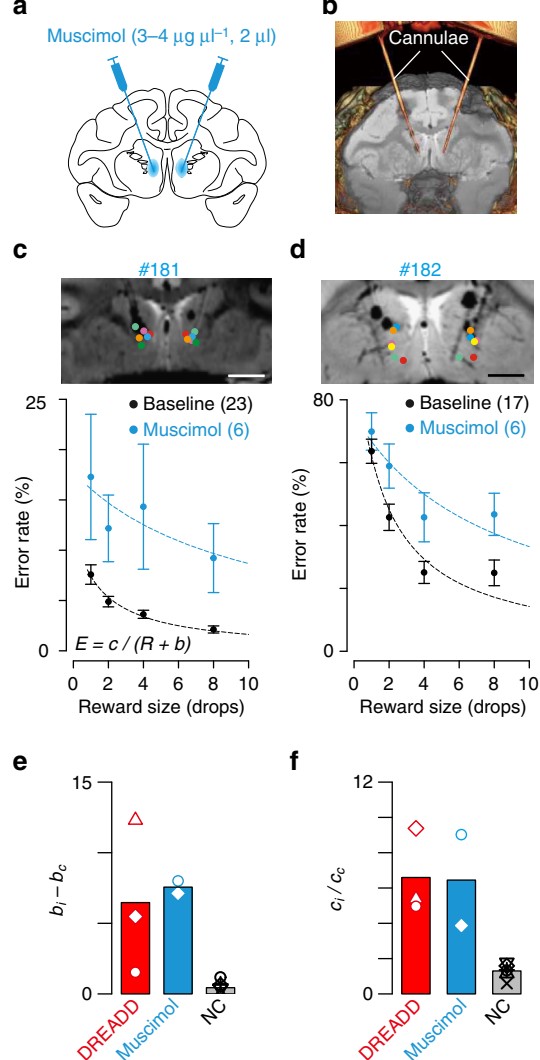

**Figure 5 | DREADD versus muscimol inactivation.** (**a**) Illustration of intended muscimol injection sites. (**b**) Identification of injection sites using CT. (**c**) (top) Location of a pair of muscimol injections for each session indicated by different coloured dots on a coronal MR image for monkey #181. Scale bar = 5 mm. (bottom) Behavioural performance for baseline (black) and inactivation of bilateral rmCD by muscimol (cyan). Error rates (mean ± s.e.m.) are shown as function of reward size. Dotted curve is the best fit of inverse function. (**d**) Muscimol injection sites and behaviour for monkey #182. (**e,f**) Comparison of treatment-induced changes in best-fit parameters of $b$ and $c$, respectively. DREADD indicates data obtained from monkeys #171, #184 (after second injection), and #190. Muscimol indicate the data obtained from monkeys #181 and #182. NC denotes data obtained from negative cases ($n = 6$) that are shown in Supplementary Fig. 4. Points indicate values derived from individual subjects, while bars indicate average across subjects. Subscripts $i$ and $c$ indicate best-fit parameters for inactivation and control condition, respectively.

but did not change the reward-size sensitivity (Fig. 5e,f, included in NC), suggesting that abnormal reward evaluation was ascribable to bilateral rmCD inactivation. Thus, chemogenetic silencing appeared to produce behavioural changes similar to pharmacological inactivation at the same location.

## Discussion

In the present work, we demonstrated that repeatedly inactivating the rmCD through the use of CNO to activate the inhibitory DREADD, hM4Di, produced a clear and reproducible loss of sensitivity to reward value in the decision-making of monkeys. The rmCD inactivation did not seem to decrease attention or drive; the reaction times and the total reward earned were unaffected. This decreased reward sensitivity is in line with the presumed function of the rmCD, a structure receiving strong input from several prefrontal cortical areas that are crucial to reward-related behaviour[1]. The decreased sensitivity to relative value during rmCD silencing is similar to that seen after bilateral removal of the OFC[10]. It has been suggested that the rmCD integrates information from the prefrontal areas[11], including the OFC, an area known to be responsible for motivational and reward-related information[12–17]. Silencing the rmCD with muscimol produced the same behavioural effect as silencing with DREADD, providing confirmation that the effect is specific to the perturbation of neuronal activity in this region, and not an artefact of the technique employed. From these experiments, and given the lack of effect when the DREADD failed to express at the intended target (*cf.* Supplementary Fig. 4), we conclude that this small region we identify as rmCD is essential for assessing predicted reward values.

Because temporary inactivation of the rmCD using the hM4Di receptor versus muscimol produced similar effects, either of the silencing methods might have been sufficient for our experiment. However, chemogenetic silencing offers several advantages over pharmacological inactivation, so we were eager to pursue its application. First, once the hM4Di-DREADD expression becomes stable (that is, 4–6 weeks following injection of the vector), local inactivation can be induced repeatedly by systemic CNO administration, whereas local pharmacological inactivation requires implanted or repeated insertion of a cannula, making it likely that there will be ongoing mechanical damage. Second, because the neurons expressing hM4Di-DREADD form a stable population, every inactivation is virtually identical to every other one, whereas the effects of local pharmacological inactivation vary session-by-session due to variation in the spread and exact dose, making it difficult to know precisely which tissue has been affected. The region expressing hM4Di-DREADD can be identified by post-mortem analysis with histochemical staining at cellular resolution. Third, with multiple viral vector injections, extensive regions of tissue can be affected[18], or, as shown here, discontinuous regions can be inactivated simultaneously via single systemic administration of the inducer. Fourth, cell-specific promoters[9] or pathway-selective expression systems[19,20] can drive expression of hM4Di-DREADD in specific neuronal subsets or pathways, enabling manipulation of a neural circuit defined by a cell type, an anatomical connection, or other markers. Finally, synaptic silencing via localized microinfusion of CNO at a site distal to the neurons expressing hM4Di-DREADD enables pathway-selective functional analysis[21].

We also report that the hM4Di receptor can be imaged *in vivo* with [11C]CLZ-PET. PET imaging provides a means to visualize the location of and measure the stability of hM4Di-DREADD expression in this subcortical region (*cf.* Supplementary Fig. 2). Knowing whether the correct region has been affected when the expression has been stabilized is of great advantage for

conducting behavioural experiments, especially in nonhuman primates, which often require the investment of considerable time, effort and other resources in each individual subject. The potential value of this approach is shown by the correlation between weak/mistargeted DREADD expression and the absence of effect on behaviour (*cf.* Supplementary Fig. 4b,c); these results indicate that PET visualization of DREADD expression would make a useful screening procedure prior to beginning a series of behavioural experiments, and especially longitudinal studies.

[11C]CLZ-PET also provides a way of measuring *in vivo* DREADD receptor occupancy by CNO. Based on the dose-occupancy curve (*cf.* Fig. 2d), we chose a dose of 3 mg kg$^{-1}$ CNO, yielding ∼60% receptor occupancy, to carry out our behavioural studies. This dose of CNO produced a repeatable behavioural modulation in reward sensitivity. At this dose there was no CLZ in cerebrospinal fluid (CSF). We did not see signs of other behavioural effects of CNO such as sedation or loss of motivation as were seen at a higher dose (10 mg kg$^{-1}$) in our previous study[18]. This latter observation is relevant because there is some evidence that in humans and guinea pigs, at least, CNO can be metabolized to CLZ, an atypical antipsychotic drug with sedative effects[22,23]. Because of the concern over side effects, other DREADD agonists such as compound 21 or Perlapine[24] should be investigated for their potential utility in primates, and perhaps in the long run in humans.

The success in imaging the hM4Di receptor with [11C]CLZ almost certainly depends on good permeability of the blood–brain barrier by CLZ and the high affinity of the hM4Di receptor for CLZ[25]. The baseline brain uptake of [11C]CLZ-PET reflects a modest affinity (∼10 nM of $K_i$) of CLZ for many innate receptors, including serotonergic, dopaminergic, adrenergic, histaminergic and muscarinic receptors[6]. Despite the resulting possibility of interference by its binding to such receptors, uptake at hM4Di-expressing regions was about 20% higher than the natural background in five monkeys, a sufficiently high signal-to-noise ratio to permit reliable detection using the reference method (*cf.* Fig. 2a). As more specific ligands are developed, the utility of PET for monitoring gene expression in the brain should improve considerably.

In summary, using the CNO-activated hM4Di-DREADD, we have revealed that the rmCD is involved in making normal judgments about the value of reward, which is largely consistent with previous anatomy and single-neuron recording studies. The mechanism by which DREADDs work, and their potential for covering a large and diffuse population of neurons, as well as the capability of verifying DREADD expression by PET imaging before undertaking functional CNO activation experiments, ought to make this system an attractive tool for application to long-term behavioural studies, and potentially to clinical settings in the future.

## Methods

**Subjects.** The subjects were 12 macaque monkeys (two cynomolgus (*Macaca fascicularis*; #127, #157), eight rhesus (*Macaca mulatta*; #152, #178, #181, #182, #183, #184 #190, #199) and two Japanese monkeys (*Macaca fuscata*; #171, #174); 4.2–9.8 kg; age 5–14 years; all were male except #174). All experimental procedures were carried out in accordance with the Guide for the Care and Use of Laboratory Animals (National Research Council of the US National Academy of Sciences) and were approved by the Animal Care and Use Committee of the National Institute of Radiological Sciences. The monkeys were kept in individual primate cages in an air-conditioned room. A standard diet, supplementary fruits/vegetables and a tablet of vitamin C (200 mg) were provided daily.

**Viral vector production.** The AAV2-CMV-hM4Di (AAV-hM4Di; Fig. 1a) and AAV2-CMV-HA-hM4Di (AAV-HA-hM4Di) vectors were produced by the helper-free triple transfection procedure and was purified by affinity chromatography (GE Healthcare). Viral titre was determined by quantitative PCR using Taq-Man technology (Life Technologies). The transfer plasmid (pAAV-CMV-hM4Di-WPRE)

was constructed by inserting hM4Di fragment and WPRE sequence into an AAV backbone plasmid (pAAV-CMV, Stratagene).

The Lenti-hSyn::hM4Di-CFP (Lenti-hM4Di-CFP) and Lenti-syn::CFP (Lenti-CFP) vectors were produced as previously described[18,26]. I293T cells (Lenti-X Invitrogen 632180, Life Technologies) were transfected with the Lenti-backbone plasmid and packaging plasmids (gift from Didier Trono; Addgene plasmids #12260 and #12259). Supernatant was replaced with Ultraculture medium (Invitrogen). The lentiviral vector containing supernatant was collected and concentrated (Beckman S28 rotor, at 22,000 r.p.m.), and aliquots were stored at − 80 °C. Virus titre was determined by quantitative PCR as described previously[9]. The transfer plasmid (hM4Di-CFP gene) was constructed by replacing the mCherry sequence with cerulean cyan fluorescent protein (CFP) in the hM4Di-mCherry sequence[27], and cloned into the Lenti-syn::GFP vector[9] by replacing the GFP sequence.

**Surgical procedures and viral vector injections.** The monkeys used for vector injection and viral vector, volume, titre and location of injection are summarized in Supplementary Table 1. Prior to surgery, overlay magnetic resonance (MR) and CT images were created using PMOD image analysis software (PMOD Technologies Ltd, Zurich, Switzerland) to estimate stereotaxic coordinates of the target brain structures. Under isoflurane general anaesthesia (1–2%), the monkeys underwent a surgical procedure to open burr holes (∼8 mm diameter) for the injection needle. Three monkeys (#171, #184 and #190) had bilateral injections into the rmCD, with 3 μl of AAV-hM4Di ($2.0 \times 10^{13}$ particles per ml) injected in each side. Two other monkeys (#182 and #183) had bilateral injections of 3 μl of AAV-HA-hM4Di ($1.8 \times 10^{13}$ particles per ml) into the rmCD on each side. Two monkeys (#127 and #157) had bilateral injections into the lateral putamen, with 30 μl of Lenti-hM4Di ($2 \times 10^9$ particles per ml) in one side and 30 μl of Lenti-CFP ($2 \times 10^9$ particles per ml) in the other. Viruses were pressure-injected by 50 or 10-μl Hamilton syringe (Model 705 RN or 701 RN, Hamilton) with a 30-gauge injection needle and a fused silica capillary (450 μm OD) to create a step about 500 μm away from the needle tip to minimize backflow. The Hamilton syringe was mounted into a motorized microinjector (IMS-10, Narishige) that was held by manipulator (Model 1460, David Kopf, Ltd.) on the stereotaxic frame. After the dura mater was opened about 5 mm, the injection needle was inserted into the brain and slowly moved down 2–3 mm beyond the target and then kept stationary for 5 min, after which it was pulled up to the target location. The injection speed was set at 0.5 μl min⁻¹. After each injection, the needle remained in situ for 15 min to minimize backflow along the needle.

For muscimol injection, two monkeys (#181 and #182) had surgery under isoflurane anaesthesia (1–2%) to implant a head-fixation device and two chambers (CRIST Instrument Company., Inc.) targeting the caudate nucleus at a 20° angle in the coronal plane.

**Drug administration.** CNO (Toronto Research) was dissolved in 2.5% of dimethyl sulfoxide (DMSO) in saline to a final volume of 6 ml. For behavioural testing, a catheter was placed in the saphenous vein while the monkey was seated in a primary chair. CNO (3 mg kg⁻¹) or the vehicle alone (2.5% DMSO) was administered at a rate of 0.2 ml s⁻¹ intravenously via the catheter 15 min before behavioural testing. For PET occupancy studies, CNO solution (doses 1, 3 and 10 mg kg⁻¹) was administrated intravenously via a saphenous vein catheter 10 min before PET imaging. For in vitro electrophysiology, CNO was diluted into 1 μM with artificial CSF. Fresh CNO solution was prepared on the day of usage.

Muscimol (Sigma) was dissolved in saline at a dose of 3 μg μl⁻¹ (#181) or 4 μg μl⁻¹ (#182). The solution (2 μl) was injected at a rate of 0.2–0.4 μl min⁻¹ through an injection-recording cannula (350 μm OD, BRC Inc. Japan) inserted stereotaxically into each hemisphere. Each cannula was held by an oil-drive manipulator (MO-97A, Narishige). Behavioural testing was started after the injection was completed. Locations of injection cannulae were visualized using CT scans after each muscimol injection session, and the locations of the tip were mapped onto MR image using PMOD.

**Radiosynthesis and PET imaging.** [¹¹C]CLZ was radiosynthesized based on the described protocol[28], and its radiochemical purity and specific radioactivity at the end of synthesis exceeded 95% and 37 GBq μmol⁻¹, respectively. PET scans were performed using a microPET Focus 220 scanner (Siemens Medical Solutions, USA), which yields a 258 mm diameter × 76 mm axial field of view (FOV) and a spatial resolution of 1.3 mm full width at half maximum at the centre of FOV[29]. The monkey was anaesthetized with isoflurane (1–2%) during all PET procedures. Emission scans were acquired for 90 min in a three-dimensional list mode after bolus injection of [¹¹C]CLZ (265–405 MBq) intravenously via a saphenous vein catheter. A transmission scan using a spiralling ⁶⁸Ge-⁶⁸Ga point source was performed for correction of attenuation before injection of the radioligands. All list-mode data were sorted into three-dimensional sinograms, which were then Fourier-rebinned into two-dimensional sinograms (frames × minutes; 5 × 1, 5 × 2, 5 × 3 and 12 × 5). Images were thereafter reconstructed with filtered back-projection using a Hanning filter cut-off at the Nyquist frequency (0.5 mm⁻¹). Volumes of interest (VOIs) were placed manually on a series of coronal slices of individual MRI. VOI for the baseline striatal reference region was located to cover

all striatal regions excluding five coronal sections centred on the injection sites (that is, needle tracks). For the time course analysis, the target and control VOIs for putaminal injections were defined as follows: target VOI: the area above 120% baseline striatum in the final image (taken at day 72 and 575 for #127 and #157, respectively); control VOI: the corresponding contralateral area. For rmCD injections, the target VOI was located in the lower half of the CD along with the needle track (cf. Fig. 1c). Each VOI was placed on the PET images to calculate the standardized uptake value (SUV), as the concentration of radioactivity in the VOI (Bq cm⁻³) × body weight (g)/injected radioactivity (Bq) averaged between 30 and 90 min after injection of the radioligand using PMOD. We then calculated the uptake ratio to the baseline striatum and expressed this as % striatal baseline.

Estimates of fractional occupancy (Occ; cf. Fig. 3d) were derived with regard to non-treatment as,

$$\mathrm{Occ} = \frac{\mathrm{URS_{NT}} - \mathrm{URS_{CNO}}}{\mathrm{URS_{NT}} - 100},$$

where $\mathrm{URS_{CNO}}$ and $\mathrm{URS_{NT}}$ indicate uptake ratio (%) of target to baseline striatum under CNO pretreatment and non-treatment conditions, respectively. The relationship between occupancy (Occ) and CNO dose ($D_{CNO}$) was modelled by the Hill equation,

$$\mathrm{Occ} = \frac{(D_{CNO})^n}{(D_{CNO})^n + (\mathrm{ED}_{50})^n},$$

where $\mathrm{ED}_{50}$ and $n$ indicate the CNO dose achieving 50% occupancy and the Hill coefficient, respectively (cf. Fig. 2d).

**Validation of PET visualization for hM4Di-DREADD expression.** To validate the PET visualization of striatal hM4Di expression, we targeted the central putamen because it has the largest volume among the three striatal subregions (putamen, caudate and ventral striatum), and the baseline uptake of [¹¹C]CLZ in these regions did not differ ($N = 3$ monkeys; one-way ANOVA, $F_{2,3} = 0.008$, $P > 0.99$). Two monkeys were injected with a lentiviral vector expressing hM4Di fused with CFP under the neuron-specific promoter synapsin (Lenti-hM4Di-CFP) in the right putamen, and a control lentiviral vector (Lenti-CFP) in the left putamen (Supplementary Fig. 2a,b). We performed a [¹¹C]CLZ-PET scan with a monkey (#157) at 82 days after injection. Focal high uptake (>6 SUV) was found in the right putamen compared with the left (Supplementary Fig. 2c). Post-mortem immunohistochemical examination showed that hM4Di-CFP-positive neurons were found at the injection site (Supplementary Fig. 2f), where high uptake was seen in [¹¹C]CLZ-PET (>120% baseline striatal uptake, Supplementary Fig. 2d,e). The high uptake of [¹¹C]CLZ at the injection site was similar from days 82 to 575 after injection (Supplementary Fig. 2g), suggesting that hM4Di expression levels were high and stable during this period. We found that increased [¹¹C]CLZ uptake had reached a plateau by day 46 in the second monkey #127 (Supplementary Fig. 2h–k). Thus, PET imaging enabled us to characterize the location and stability of DREADD expression in the striatum.

**MR imaging and CT scans.** MR imaging and CT scans were performed under general anaesthesia (intravenously continuous infusion of propofol 0.2–0.6 mg kg⁻¹ per minute, or pentobarbital sodium 15–30 mg kg⁻¹, i.v.). MR images were obtained with a 7T 40 cm-bore Avance-I system (KOBELCO/Bruker Biospin) or a 1.5T Philips GYROSCAN Interna. The following sequences were used for 7T: three or four sets of FLASH sequence (TR = 660.0 ms, TE = 13.37 ms, slice thickness = 1.1 mm, matrix = 512 × 512, flip angle = 20°, FOV = 150 mm, NEX = 1, number of slices 24). The sequences for 1.5T: three-dimensional T1-weighted imaging (TR = 30 ms, TE = 6 ms, matrix = 512 × 512, FOV = 256 mm, slice thickness = 1.0 mm without slice gap, number of slices = 60). CT scans were obtained using cone-beam CT system (Accuitomo170, J.MORITA CO., Japan), which operated with tube voltage = 90 kVp, tube current = 5 mA, exposure time = 17.5 s, FOV = 140 mm diameter × 100 mm height, voxel size 0.25 × 0.25 × 0.5 mm³, and a grey intensity of 16 bits.

**Behavioural testing and analysis.** A total of seven monkeys were used for behavioural testing. All monkeys had been trained to perform colour discrimination trials in the cued multi-trial reward schedule task for more than 3 months. Task control and data acquisition were performed using the REX program. The use of an automated system eliminated the need for experimenters to be blind to treatment. In this study, we used a reward-size task as described previously[8] (Fig. 1c). A monkey initiated a trial by touching the bar. A visual cue and a red target sequentially appeared. After a variable interval, the red target turned green. If the monkey released the bar between 200 and 1,000 ms, a reward of 1, 2, 4 or 8 drops of liquid reward (1 drop = ∼0.1 ml) was delivered immediately after the signal turned to blue. If the monkey released too early (∼200 ms after the green target appeared) or failed to respond within 1 s after the green target appeared, we regarded the trial as an 'error trial'; the trial was aborted and the trial was repeated after the 1s inter-trial interval. The visual cue presented at the beginning of the trial indicated the number of drops for the reward (Fig. 1d). In this task, our behavioural measurement for the expected outcome value was the proportion of error trials. Since the monkeys were able to perform the task correctly in nearly

every trial when the reward size was not assigned, an error trial is regarded as a trial in which the monkeys are not sufficiently motivated to release the bar correctly[8]. Before each testing session, the monkeys were subject to ∼22 h of water restriction without any behavioural testing. Each testing session continued for 100 min, although most monkeys had reached satiety and ceased to initiate trials by 90 min.

Three and two monkeys received bilateral injection of AAV-hM4Di vector and AAV-HA-hM4Di vector, respectively. Two monkeys were given muscimol injections, and two monkeys served as non-operated controls. All monkeys had been trained with the reward-size task for at least 3 weeks. For vector-injection monkeys, pre-vector baseline data were collected for 15–23 sessions, and post-vector data collection began 15 days after the vector injection. CNO and vehicle treatment were tested no more than once per week. For muscimol-injection monkeys, after recovery from the surgery of head post and chamber (at least 2 weeks), muscimol injection was performed once per week. Non-treatment baseline data were collected in the same week of the injection except for the day following the injection. Similarly, behavioural data for non-operated controls were collected on the day of CNO treatment (CNO; four sessions per monkey) and on the day without treatments (baseline).

Error rate for each reward size was calculated for each daily session. We used the error rates to estimate the level of motivation, as the error rates of these tasks ($E$) were inversely related to the value for action[8]. In the reward-size task, we used an inverse function,

$$E = \frac{c}{R + b},$$

where $E$ is error rate, $R$ is reward size, $c$ and $b$ are constants. We fitted the inverse function to the data by sum-of-squares minimization. The coefficient of determination ($R^2$) is reported as a measure of the goodness-of-fit.

We also performed repeated-measures ANOVA to test the effect of treatment × reward size or treatment × reward size × time from injection on error rates. Tukey's HSD multiple comparison tests were conducted as a *post hoc* test. The arcsine transformation was used to normalize the distribution of binomial data (that is, error rates) before statistical analysis[30].

Details on data acquisition and analysis for histology, bioavailability of CNO/CLZ and electrophysiology are presented in the Supplementary Methods.

**Data availability.** The data that support the findings of this study are available from the corresponding author upon reasonable request.

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

## Acknowledgements

We thank J. Kamei, R. Yamaguchi, Y. Matsuda, Y. Toyoda, M. Kurokawa, Y. Iwasawa, S. Shibata, N. Nitta, Y. Ozawa, M. Fujiwara, K. Nagaya and V. Der Minassian for their technical assistance, Dr K. Takemoto for his technical suggestions, and the staff of Department of Radiopharmaceuticals Development, NIRS, for support with radiosynthesis. This study was supported by PRESTO/JST and JSPS KAKENHI JP25135736, JP26282221 and JP15H05917 (to T.M.), JP15H01419 (to T.H.), JP25640011 (to M.T.), by the IRP-NIMH/NIH (B.J.R., W.L. and M.A.G.E.), by Brain/MINDS and the SRPBS from AMED, by the cooperative research program at PRI, Kyoto University and by National Bio-Resource Project 'Japanese Monkeys' of MEXT, Japan.

## Author contributions

Y.N. and T.M. designed the study. Y.N., B.J., Y.Ka., Y.H., Y.Ki., T.H., A.F., I.A., T.S., M.H., B.J.R. and T.M. performed and analysed the imaging studies. K.K. and M.-R.Z. performed radiosynthesis. K.I. and W.L. produced the virus vectors. Y.N., W.L., Y.H., Y.Ka., B.J.R. and T.M. performed the surgery. E.K., Y.H. and T.M. performed and analysed the behavioural experiments. Y.N., K.I., M.T. and T.M. analysed the histological data. H.K., B.J., M.H. and T.M. performed and analysed the *in vitro* experiments. A.O.-N. and B.J. analysed the serum and CSF samples. All authors contributed to writing and editing the manuscript.

## Additional information

**Competing financial interests:** Y.N., B.J., T.S., M.H., and T.M. are named as inventors on a patent application in Japan claiming subject matter related to the results described in this paper. The remaining authors declare no competing financial interests.

