## [Peer Review File · Nature Communications]

Reviewers' comments:

Reviewer #1 (Remarks to the Author):

This is a potentially interesting paper which demonstrates the potential utility of combining PET imaging of chemogenetic GPCRs with behavior in nonhuman primates. It is likely to be of interest to a wide variety of neuroscientists--in particular those who use primates--and potentially has translational value as well.

The authors essentially show that hM4Di reliably apparently induces silencing of a similar valence and magnitude as muscimol and suggest advantages for a chemogenetic approach rather than muscimol injections. They also show that hM4Di can be apparently visualized via ¹¹C-CLZ PET imaging which makes this potentially highly useful from both a basic science and logistical perspective.

The authors mention potential issues with CNO in primates and it would be useful to mention that other CNO analogues have been described in the literature and mention their potential advantages/disadvantages over CNO.

SPECIFIC CONCERNS:

1. If the authors have validated chemogenetic neuronal silencing via slice ePhys than those data should be in the main text and a main figure. If the authors have not done this essential control they would need to do it prior to considering this paper for publication.
2. Have the authors performed simultaneous fMRI measurements in the same regions?

Reviewer #2 (Remarks to the Author):

In this study, Nagai et al. pioneered a DREADD-based method to examine rostro-medial caudate's roles in reward value processing in monkeys. This is to my knowledge the first demonstration of the usefulness of DREADD techniques in studying the neural mechanisms of cognition in non-human primates. The application of PET imaging to measure the efficacy of DREADD expression in vivo demonstrates additional, significant advantages over traditional pharmacological/lesioning techniques. The techniques are novel, solid with rigorous controls and have the potential to open up new directions of research in the field. Below I describe some suggestions/comments that are intended to improve the clarity of the paper.

Major:

- It seems to me that the manuscript's foremost contribution is the potentially powerful new techniques. The effects with rmCD serve as an indicator of the effectiveness of the new

technique. However, the Abstract and Introduction led me to assume initially that the main point of the manuscript is about the causal roles of rmCD. There was already a previous study demonstrating a causal relationship between caudate activity and reward-modulated behavior, using unilateral injection of dopamine antagonist (Nakamura and Hikosaka 2006 J Neurosci). I was also not sure why a simple muscimol injection experiment, just like what the authors described later in the manuscript, would not be sufficient. The authors addressed the latter question in Discussion. But these concerns unnecessarily detract from the main achievement of the study. I would thus suggest a re-structuring of the manuscript to highlight the motivation for developing the new techniques early on.

- Pg. 6, near the bottom, "For all three monkeys, the reaction times were not affected by CNO administration (two-way ANOVA, main effect of treatment, $F_{1, 134} = 0.12$, $p = 0.72$). The total reward earned was not affected by CNO (ANOVA, main effect of treatment, $F_{1, 32} = 3.1$, $p = 0.09$)." I am puzzled by these results. Since error trials are defined as too-fast or too-slow responses, presumably increased error rates must reflect changes in the reaction time distribution? For the second finding, given the higher error rates, the only way total reward earned did not change is if the monkeys performed more trials. Could silencing rmCD increase motivational drive?

- Fig 2d can be improved for clarity. For example, as is, it appears that error rates did not change with reward size for the vehicle, 0-20 min condition. Given that there are only 5 timing conditions, it would be better in my opinion to just plot 5 panels of error rate vs. reward size curves (similar to Fig 2b) for CNO/vehicle pairs, side-by-side for different timing conditions, instead of the surface plot.

- Pg. 8 line 6 from bottom, "Unilateral inactivation of the rmCD did not change the overall error rates (main effect of treatment, $p = 0.065$; Supplementary Fig. 4d)". This statement is not well supported, given that the data came from only one monkey and there is a clear trend.

Minor:

- Are the results in supplementary Fig1 from caudate neurons?

- Suppl Fig 2d looks more like an MRI image with an overlay, not a PET image? What does the overlay mean? What's different between (d) and (e)?

- Fig 2C needs definitions of P, C, PC and V for the tick marks.

- Supplementary Fig. 3: the best-fit Hill coefficient should be given.

- Pg 6 first line. "Fig. 4b" here should be Fig 2d?

- Pg 6. "The CNO treatment increased the overall error rates in these monkeys (main effect of treatment, $p < 0.01$, Fig. 3c, d)." Were there occupancy data from these two monkeys? If so, they should be included.

- Pg. 7, line 5-6. "Explain" implies mechanistic knowledge. Perhaps it is more appropriate to state that the equation "describes" the observed effect.

Reviewer #3 (Remarks to the Author):

The manuscript examines the use of PET-imaging guided chemogenetics to make perturbations of caudate nucleus neurons in the monkey. This is a sensible approach that appears to work well, and I support publication of the paper. One additional piece of information would be useful, which is examination of potential tachyphylaxis in both the behavioral perturbation and receptor availability with successive doses of CNO.

Reviewer #4 (Remarks to the Author):

In the present manuscript, Nagai and collaborators did investigate the impact of chemogenetic silencing of striatal neurons (by DREADD approach) on reward evaluation in non-human primates.

After in vitro characterization of their viral construct (hM4Di, the modified muscarinic receptor), the authors did bilaterally inject it within the rostro-medial part of the primate caudate nucleus and did control longitudinally its expression level by PET imaging using the radioligand ¹¹C-Clozapine. They have then looked for the dose of clozapine-N-oxide (CNO) to use, as clozapine-N-oxide binds to and activates hM4Di (and therefore to suppress neuronal activity of neurons expressing hM4Di). Once the right conditions were found, the authors did test the impact of such inactivation on reward value evaluation by using a reward-size task, which they have used extensively in the past. They show that inactivating the rostro-medial part of the caudate nucleus by systemic administration of the drug inducer leads to a specific increase of error rates (without any change in reaction time or total reward earned; but these data are not shown). The task's performance returned to baseline levels the day after, showing the reversibility of CNO treatment. In the last part of the manuscript, the authors did compare their DREADD results with bilateral injections of muscimol (a GABA_A agonist) within the same region of the caudate nucleus. Muscimol also led to an increase of error rates and loss of sensitivity to reward value, as expected.

This study is well designed, written, novel and complete. The authors did provide several control experiments (from in vitro to in vivo), which strongly reinforce the impact of their results. Besides this methodological point of view, the study doesn't provide new advances on the function of the caudate nucleus but the study is still very important to the research community as it combines for the first time PET imaging with DREADDs approach in the monkey and demonstrates the critical role of the caudate in reward value processing.

I have a few remaining comments:

The authors have been using three different monkey species (please specify male or female) (cynomolgus, rhesus and japanese) and different viral constructs (AAV2-CMV-hM4Di; AAV2-

CMV-HA-hM4Di; Lenti-hSyn-hM4Di; Lenti-hSyn-CFP) for the experiments. I would suggest to add a table summarizing this information (monkey, viral construct, volume and titer, type of injection, type of experiment, etc). The relevant experiments are often performed on rhesus or japanese monkeys with AAV2-CMV-hM4Di, and the control experiments are performed on cynomolgus monkeys with lenti-hM4Di-CFP. The authors have also been using AAV-HA-hm4Di. Why? Lentivirus and adeno-associated virus are supposed to have different transfection properties, isn't it? Lentiviral injections were further performed within the putamen and not the caudate, the region of interest in the paper. Please explain why. Could you also explain why the muscimol experiments were not performed on the monkeys with AAV-CMV-hM4Di?

The discussion is quite methodological.

Minor comments

-Figure 1 (and methods): How is made the intravenous CNO injection? For PET imaging, the monkey is under anesthesia, but the monkey is awake during the task.

-B and c parameters are not shown on Figure 2b

-Figure 3A: this is quite a large area that has been transfected now after the second injection.

-Legend of Supplementary Figure 1: remove "bus" and replace by "bath"

-The title of Supplementary Figure 2 is not informative enough

-Legend of Supplementary Figure 2: What is CFP?

-Supplementary Figure 3: The authors did illustrate monkey 157 instead of monkey 171. Why?

This experiment of occupancy of hM4Di is very important. I would suggest to move this part of the figure at the beginning because this is a key experiment, which determined the in vivo CNO dose to use. Why did the authors illustrate displacement of binding at the level of the putamen and not at the level of the caudate itself?

-Supplementary Figure 4: please modify the title as monkey 184 is not a control monkey.

-page 6, first line: This is not Figure 4b but Figure 2b.

-page 6: please add a space before (Fig. 2d)

-methods, subjects: please indicate male or female

-methods, drug administrations: in which vein is CNO injected? Please specify for both PET and behavioral approaches.

Reviewers' comments:

Reviewer #1 (Remarks to the Author):

This is a potentially interesting paper which demonstrates the potential utility of combining PET imaging of chemogenetic GPCRs with behavior in nonhuman primates. It is likely to be of interest to a wide variety of neuroscientists--in particular those who use primates--and potentially has translational value as well.

The authors essentially show that hM4Di reliably apparently induces silencing of a similar valence and magnitude as muscimol and suggest advantages for a chemogenetic approach rather than muscimol injections. They also show that hM4Di can be apparently visualized via ¹¹C-CLZ PET imaging which makes this potentially highly useful from both a basic science and logistical perspective.

R: We appreciate the reviewer's favorable opinion.

The authors mention potential issues with CNO in primates and it would be useful to mention that other CNO analogues have been described in the literature and mention their potential advantages/disadvantages over CNO.

R: We thank the reviewer for suggesting that we mention CNO analogues. We have added the following discussion regarding potential issues with CNO metabolites and mentioning using CNO analogues.

"[¹¹C]CLZ-PET also provides a way of measuring *in vivo* DREADD receptor occupancy by CNO. Based on the dose-occupancy curve (cf. Fig. 2d), we chose a dose of 3 mg kg⁻¹ CNO, yielding approximately 60% receptor occupancy, to carry out our behavioral studies. This dose of CNO produced a repeatable behavioral modulation in reward sensitivity. At this dose there was no CLZ in cerebrospinal fluid (CSF). We did not see signs of other behavioral effects of CNO such as sedation or loss of motivation as were seen at a higher dose (10 mg kg⁻¹) in our previous study¹⁸. This latter observation is relevant because there is some evidence that in humans and guinea pigs, at least, CNO can be metabolized to CLZ, an atypical antipsychotic drug with sedative effects^{22, 23}. Because of the concern over side effects, other DREADD agonists such as compound 21 or Perlazine²⁴ should be investigated for their potential utility in primates, and perhaps in the long run in humans." (pg. 12, lines 6-17)

SPECIFIC CONCERNS:

1. *If the authors have validated chemogenetic neuronal silencing via slice ePhys than those data should be in the main text and a main figure. If the authors have not done this essential control they would need to do it prior to considering this paper for publication.*

R: Chemogenetic neuronal silencing was validated via an *in vitro* electrophysiological assay using cultured neurons expressing hM4Di after being infected using our AAV construct (Supplementary Fig. 1). We believe that this substitutes for what the reviewer requested.

We refer to the result in the main text; "We verified that activation of the hM4Di receptor with CNO caused neuronal silencing *in vitro* (Supplementary Fig. 1)." (pg 5, line 9). We appreciate the suggestion that the result be in the main figure. Because this replicates a finding from our previous study, with the difference being the viral vector (lentivirus vs. AAV), we believe that it is appropriate for inclusion as a Supplementary Figure.

Although a limited number of studies report electrophysiological examination in monkey brain slice (e.g., Alexander et al., Neuroscience, 2006), it is notoriously difficult to prepare such material. Furthermore, this is not a standard preparation. It would be difficult on ethical grounds to defend sacrificing our rhesus monkeys to carry out such experiments.

We believe that we have presented a convincing chain of evidence that we achieved neuronal silencing in our behaving monkeys: 1) Chemogenetic neuronal silencing was validated via an *in vitro* electrophysiological assay using cultured neurons expressing hM4Di (Supplementary Fig. 1). 2) Using the same AAV vector, hM4Di was expressed in bilateral rmCD in three monkeys (cf. Fig. 2a). 3) CNO dose of 3 mg/kg specifically blocked [¹¹C]CLZ binding to the striatal hM4Di and yielded about 60% occupancy without conversion to CLZ (Fig. 2 and Fig S3). 4) CNO administration specifically altered reward evaluation in 3 monkeys expressing hM4Di in bilateral rmCD (Fig. 3 and 4). Given this chain of evidence, we feel strongly that recording from brain slices is not an essential control experiment for our study or future behavioral studies using DREADDs in macaque monkeys.

[redacted]

Reviewer #2 (Remarks to the Author):

In this study, Nagai et al. pioneered a DREADD-based method to examine rostro-medial caudate's roles in reward value processing in monkeys. This is to my knowledge the first demonstration of the usefulness of DREADD techniques in studying the neural mechanisms of cognition in non-human primates. The application of PET imaging to measure the efficacy of DREADD expression in vivo demonstrates additional, significant advantages over traditional pharmacological/lesioning techniques. The techniques are novel, solid with rigorous controls and have the potential to open up new directions of research in the field. Below I describe some suggestions/comments that are intended to improve the clarity of the paper.

R: We appreciate the reviewer's favorable opinion.

Major:

- It seems to me that the manuscript's foremost contribution is the potentially powerful new techniques. The effects with rmCD serve as an indicator of the effectiveness of the new technique. However, the Abstract and Introduction led me to assume initially that the main point of the manuscript is about the causal roles of rmCD. There was already a previous study demonstrating a causal relationship between caudate activity and reward-modulated behavior, using unilateral injection of dopamine antagonist (Nakamura and Hikosaka 2006 J Neurosci). I was also not sure why a simple muscimol injection experiment, just like what the authors described later in the manuscript, would not be sufficient. The authors addressed the latter question in Discussion. But these concerns unnecessarily detract from the main achievement of the study. I would thus suggest a re-structuring of the manuscript to highlight the motivation for developing the new techniques early on.

R: We are sympathetic to the reviewer's point of view. However, a technique becomes most valuable when it can be applied to bring new insights, here about caudate function. The reviewer's comments overlook the level of current understanding, by which not all sites in the caudate are equal. We believe that there has been no previous work, certainly none that addresses the intimate connection between causality and neuronal activation, that shows that the rostromedial caudate is essential for normal reward evaluation. The Nakamura-Hikosaka study was largely centered more caudally and more laterally, and as was clear in their manuscript, they had trouble identifying a connection between neuronal activity and the likelihood of stimulation at the site to have an effect. Our result with the one monkey where DREADD activation failed makes it likely that a specific locus in the rostromedial caudate is where the inactivation is effective.

Without PET, though, we would have had a difficult time reaching these conclusions.

- Pg. 6, near the bottom, "For all three monkeys, the reaction times were not affected by CNO administration (two-way ANOVA, main effect of treatment, $F_{1, 134} = 0.12, p = 0.72$). The total reward earned was not affected by CNO (ANOVA, main effect of treatment, $F_{1, 32} = 3.1, p = 0.09$)." I am

puzzled by these results. Since error trials are defined as too-fast or too-slow responses, presumably increased error rates must reflect changes in the reaction time distribution? For the second finding, given the higher error rates, the only way total reward earned did not change is if the monkeys performed more trials. Could silencing rmCD increase motivational drive?

R: Reaction times were measured for correct responses only (text has been added to the 'Results' section to make this clear; pg. 7, line 7). We might expect to see a slowing of reaction times if attention or motivation was influenced by rmCD silencing, but the consistency of the reaction time distribution in the face of increased errors is what leads us to conclude that rmCD silencing caused a specific loss of sensitivity to the relative reward value.

Monkeys generally reached satiety within the 100 min testing limit (text modified to include statement to this effect; pg. 7, line 8) – thus the lack of effect of CNO on total reward earned indicates that the internal threshold for effort and motivational drive have *not* been affected.

- Fig 2d can be improved for clarity. For example, as is, it appears that error rates did not change with reward size for the vehicle, 0-20 min condition. Given that there are only 5 timing conditions, it would be better in my opinion to just plot 5 panels of error rate vs. reward size curves (similar to Fig 2b) for CNO/vehicle pairs, side-by-side for different timing conditions, instead of the surface plot.

R: We have replaced Fig 2d with 5 panels of error rate vs. reward size (Fig. 3d).

- Pg. 8 line 6 from bottom, "Unilateral inactivation of the rmCD did not change the overall error rates (main effect of treatment, $p = 0.065$; Supplementary Fig. 4d)". This statement is not well supported, given that the data came from only one monkey and there is a clear trend.

R: We rewrote the sentence as follows. "Unilateral inactivation of the rmCD produced a trend toward increased overall error rates (main effect of treatment, $p = 0.065$; Supplementary Fig. 4d), but did not change the reward-size sensitivity (Fig. 5e, f, included in NC),..." (pg. 9, line 9).

Minor:

- Are the results in supplementary Fig1 from caudate neurons?

R: The data were obtained from primary cultured neurons derived from whole brain. This is now clarified in the legend.

- Suppl Fig 2d looks more like an MRI image with an overlay, not a PET image? What does the overlay mean? What's different between (d) and (e)?

R: Supplementary Fig. 2d and e shows the focal uptake in coronal [¹¹C]CLZ-PET images overlaid on the MR images. PET images are shown only for the area over 120% striatal baseline. (d) and (e) show PET images taken at day 82 and day 575 after vector injection, respectively. They are clarified in the legends in Suppl Fig 2d.

- Fig 2C needs definitions of P, C, PC and V for the tick marks.

R: P, C, PC, and V indicate post-vector, CNO, post-CNO, and vehicle, respectively. These definitions are stated in the legend for Fig 2b.

- *Supplementary Fig. 3: the best-fit Hill coefficient should be given.*

R: According to the reviewer's suggestion, we used a Hill function with variable coefficient for approximation. We provide the best-fit coefficient, $n = 0.64$ in Figure 2d, which we have moved from Supplementary Fig. 3 according to a comment by Reviewer #4.

- *Pg 6 first line. "Fig. 4b" here should be Fig 2d?*

R: Corrected. Now it corresponds to Fig. 3a (pg. 6, line 14). Thank you.

- *Pg 6. "The CNO treatment increased the overall error rates in these monkeys (main effect of treatment, $p < 0.01$, Fig. 3c, d)." Were there occupancy data from these two monkeys? If so, they should be included.*

R: The occupancy data are available only for monkeys #171 and #157.

- *Pg. 7, line 5-6. "Explain" implies mechanistic knowledge. Perhaps it is more appropriate to state that the equation "describes" the observed effect.*

R: We have corrected this (pg. 7, last line). Thank you.

Reviewer #3 (Remarks to the Author):

The manuscript examines the use of PET-imaging guided chemogenetics to make perturbations of caudate nucleus neurons in the monkey. This is a sensible approach that appears to work well, and I support publication of the paper.

R: We appreciate the reviewer's favorable opinion.

One additional piece of information would be useful, which is examination of potential tachyphylaxis is both the behavioral perturbation and receptor availability with successive doses of CNO.

R: We agree with the reviewer's comment on the importance of examination of potential tachyphylaxis. It would be important information in successive chemogenetic silencing, which is a very powerful tool to examine specific functions such as learning, in which longitudinal PET monitoring is also valuable. We admire and are very grateful for the reviewer's deep thoughts for trying to increase the impact of our study. This study focused on basic technical advantage of DREADD/CNO such as temporal and repetitive silencing. Successful silencing is somewhat beyond the current status of DREADD technology in monkeys. Therefore, we would like to examine potential tachyphylaxis in future study.

Reviewer #4 (Remarks to the Author):

In the present manuscript, Nagai and collaborators did investigate the impact of chemogenetic silencing of striatal neurons (by DREADD approach) on reward evaluation in non-human primates.

After *in vitro* characterization of their viral construct (hM4Di, the modified muscarinic receptor), the authors did bilaterally inject it within the rostral-medial part of the primate caudate nucleus and did control longitudinally its expression level by PET imaging using the radioligand ¹¹C-Clozapine. They have then looked for the dose of clozapine-N-oxide (CNO) to use, as clozapine-N-oxide binds to and activates hM4Di (and therefore to suppress neuronal activity of neurons expressing hM4Di). Once the right conditions were found, the authors did test the impact of such inactivation on reward value evaluation by using a reward-size task, which they have used extensively in the past. They show that inactivating the rostral-medial part of the caudate nucleus by systemic administration of the drug inducer leads to a specific increase of error rates (without any change in reaction time or total reward earned; but these data are not shown). The task's performance returned to baseline levels the day after, showing the reversibility of CNO treatment. In the last part of the manuscript, the authors did compare their DREADD results with bilateral injections of muscimol (a GABA_A agonist) within the same region of the caudate nucleus. Muscimol also led to an increase of error rates and loss of sensitivity to reward value, as expected.

This study is well designed, written, novel and complete. The authors did provide several control experiments (from *in vitro* to *in vivo*), which strongly reinforce the impact of their results. Besides this methodological point of view, the study doesn't provide new advances on the function of the caudate nucleus but the study is still very important to the research community as it combines for the first time PET imaging with DREADDs approach in the monkey and demonstrates the critical role of the caudate in reward value processing.

R: We appreciate the reviewer's favorable opinion.

I have a few remaining comments:

The authors have been using three different monkey species (please specify male or female) (cynomolgus, rhesus and japanese) and different viral constructs (AAV2-CMV-hM4Di; AAV2-CMV-HA-hM4Di; Lenti-hSyn-hM4Di; Lenti-hSyn-CFP) for the experiments. I would suggest to add a table summarizing this information (monkey, viral construct, volume and titer, type of injection, type of experiment, etc). The relevant experiments are often performed on rhesus or japanese monkeys with AAV2-CMV-hM4Di, and the control experiments are performed on cynomolgus monkeys with lenti-hM4Di-CFP.

R: We have added Supplementary Table 1 summarizing the information concerning monkey species, viral constructs, volume, titer, and location of viral vector injection.

The authors have also been using AAV-HA-hm4Di. Why?

R: We intended to examine the location of DREADD expression by *in vitro* immunohistochemistry. However, AAV-HA-hM4Di vector resulted in weaker expression *in vivo*, and CNO treatment did not alter reward-size sensitivity. We reported these two monkeys as negative results.

Lentivirus and adeno-associated virus are supposed to have different transfection properties, isn't it? Lentiviral injections were further performed within the putamen and not the caudate, the region of interest in the paper. Please explain why.

R: Yes, these two types of viral vectors have different transfection properties. We used the lentiviral vector to validate *in vivo* visualization of striatal DREADD expression because it expresses hM4Di with fused tag protein CFP that can be visualized *in vitro*. For this purpose, we selected the putamen as target region, as a large volume of tissue expressing DREADD is desirable for detection, quantification and occupancy measurement. We added the explanation in the "Validation of PET visualization for hM4Di-DREADD expression" section in Methods.

Could you also explain why the muscimol experiments were not performed on the monkeys with AAV-CMV-hM4Di

R: We did not have any particular reason for this. Chemogenetic and chemical inactivation studies were performed in parallel. Behavioral results can be comparable.

The discussion is quite methodological.

R: Given the novelty of the DREADD approach with PET imaging in monkeys, careful arguments regarding the methods used this study are essential. We believe that it would be of major interest for the reader as well.

Minor comments

-Figure 1 (and methods): How is made the intravenous CNO injection? For PET imaging, the monkey is under anesthesia, but the monkey is awake during the task.

R: For behavioral testing, a catheter was placed in the saphenous vein, while the monkey was seated in a primate chair. CNO was administered via the catheter. For PET imaging a catheter was placed in the same way. We describe this in “Drug administration” and “Radiosynthesis and PET imaging” section in Methods.

-B and c parameters are not shown on Figure 2b

R: We provide parameters b and c for each treatment condition in Fig. 3b and c, respectively.

-Figure 3A: this is quite a large area that has been transfected now after the second injection.

R: The second injection unintentionally resulted in the hM4Di-positive area covering the ventral part of the striatum on the left side. In our experience, this type of misplacement or spread of virus vector occurs in some cases, and is usually confirmed by post-mortem analysis. As we stated in Discussion, our PET measurement is a powerful tool for determining this *in vivo*. Considering PET imaging with other monkeys as well as those with chemical silencing, we concluded that the behavioral alteration is due to silencing of rmCD.

-Legend of Supplementary Figure 1: remove "bus" and replace by "bath"

R: Corrected. Thank you.

-The title of Supplementary Figure 2 is not informative enough

R: We modified the title of Supplementary Fig. 2 to "Validation of PET visualization for hM4Di-DREADD expression".

-Legend of Supplementary Figure 2: What is CFP?

R: CFP stands for cyan fluorescent protein. We have added the definition in the legend as follows.

"(a) Structure of the lentiviral vector expressing a fusion protein [hM4Di with cyan fluorescent protein (CFP)] under a neuron-specific promoter synapsin (hSyn)."

-Supplementary Figure 3: The authors did illustrate monkey 157 instead of monkey 171. Why? This experiment of occupancy of hM4Di is very important. I would suggest to move this part of the figure at the beginning because this is a key experiment, which determined the in vivo CNO dose to use. Why did the authors illustrate displacement of binding at the level of the putamen and not at the level of the caudate itself?

R: According to the reviewer's suggestion, we have moved the figures for the occupancy studies to the main figure (Fig. 2c-d). We measured occupancy for three doses in #157 and one in #171. Now we show images for both monkeys. We also modified the text accordingly (pg. 6, top line). For occupancy study, a larger volume of DREADD-positive tissue has an advantage in accuracy for PET measurement. We mention this in Methods.

-Supplementary Figure 4: please modify the title as monkey 184 is not a control monkey.

R: We have modified the title of Supplementary Fig. 4 to "Absence of behavioral effect of CNO in monkeys with no/weak or mislocalized DREADD expression, and unilateral chemical inactivation."

-page 6, first line: This is not Figure 4b but Figure 2b.

R: Corrected. Now it corresponds to Fig. 3a (pg. 6, line 14). Thank you.

-page 6: please add a space before (Fig. 2d)

R: We added a space. Thank you.

-methods, subjects: please indicate male or female

R: All monkeys were male except for #174. We described this in Subjects section.

-methods, drug administrations: in which vein is CNO injected? Please specify for both PET and behavioral approaches.

R: CNO was administrated intravenously via a saphenous vein catheter in both PET and behavioral experiments. We describe this in Methods.

Reviewers' comments:

Reviewer #1 (Remarks to the Author):

The authors have fully addressed my concerns and I endorse publication in Nature Communications.

Reviewer #2 (Remarks to the Author):

The revision addressed all but one of my earlier concerns:

- Pg. 6, near the bottom, "For all three monkeys, the reaction times were not affected by CNO administration (two-way ANOVA, main effect of treatment, $F_{1, 134} = 0.12$, $p = 0.72$). The total reward earned was not affected by CNO (ANOVA, main effect of treatment, $F_{1, 32} = 3.1$, $p = 0.09$)." I am puzzled by these results. Since error trials are defined as too-fast or too-slow responses, presumably increased error rates must reflect changes in the reaction time distribution? For the second finding, given the higher error rates, the only way total reward earned did not change is if the monkeys performed more trials. Could silencing rmCD increase motivational drive?

Let me try again. Here is my understanding:

T: duration of a session

T_trial: average trial duration, including RT and other task intervals

total reward earned = (%correct * T / T_trial) * (average reward size from a correct trial).

The authors reported that RT (therefore T_trial) and total reward did not change with CNO. Because monkeys had to repeat error trials and cannot avoid small reward trials, the average reward size should be the same. Then with decreased %correct (i.e., higher error rate), T (# trials performed) must have increased, which might reflect a change in general motivation.

Reviewer #3 (Remarks to the Author):

I maintain my favorable view of the manuscript, however I do not agree with the authors response to decline my request to use PET imaging to examine DREADD receptor availability to repeated dosing. They responded: "This study focused on basic technical advantage of DREADD/CNO such as temporal and repetitive silencing." This is exactly right, and their method is best suited to provide insight into the core question of potential receptor availability, and it is essential information to evaluate the technical advantage of DREADD/CNO in repetitive silencing. They should perform PET imaging after successive

days of CNO administration.

Their other comment: "Successful silencing is somewhat beyond the current status of DREADD technology in monkeys." seems in contradiction to their own result, so clarification on this comment is essential.

Reviewer #4 (Remarks to the Author):

All my comments have been addressed.

Reviewer #2 (Remarks to the Author):

The revision addressed all but one of my earlier concerns:

- Pg. 6, near the bottom, "For all three monkeys, the reaction times were not affected by CNO administration (two-way ANOVA, main effect of treatment, $F_{1, 134} = 0.12, p = 0.72$). The total reward earned was not affected by CNO (ANOVA, main effect of treatment, $F_{1, 32} = 3.1, p = 0.09$)." I am puzzled by these results. Since error trials are defined as too-fast or too-slow responses, presumably increased error rates must reflect changes in the reaction time distribution? For the second finding, given the higher error rates, the only way total reward earned did not change is if the monkeys performed more trials. Could silencing rmCD increase motivational drive?

Let me try again. Here is my understanding:

T: duration of a session

T_{trial}: average trial duration, including RT and other task intervals

*total reward earned = (%correct * T / T_{trial}) * (average reward size from a correct trial).*

The authors reported that RT (therefore T_{trial}) and total reward did not change with CNO. Because monkeys had to repeat error trials and cannot avoid small reward trials, the average reward size should be the same. Then with decreased %correct (i.e., higher error rate), T (# trials performed) must have increased, which might reflect a change in general motivation.

R: The reviewer surmises that total trials performed increased after CNO. However, the total trials performed were slightly, but not significantly, less in the CNO treatment session (ANOVA, main effect of treatment, $F_{1, 32} = 1.8, p = 0.18$). The number of correct trials (~reward earned) also slightly, but not significantly, decreased with CNO treatment. The amount of decrease was larger in correct trials than in total trials, and thereby we observed that the error rate increased without increase in total trial numbers.

To report the statistical results of the total reward earned more adequately, we rewrote the sentence; "The total reward earned was slightly, not significantly, less in the CNO treatment sessions (ANOVA, main effect of treatment, $F_{1, 32} = 3.1, p = 0.09$)." (page 7, line 10)

In our task, the error rate is the most reliable behavioral measurement, which reflects both reward size and satiation. The satiation effect is modeled by reward accumulation: the monkeys' error rates increased as total reward accumulation increased (Minamimoto et al., 2009). Since total reward earned did not change significantly, it appears that rmCD silencing does not change motivational drive.

Reviewer #3 (Remarks to the Author):

I maintain my favorable view of the manuscript, however I do not agree with the authors response to decline my request to use PET imaging to examine DREADD receptor availability to repeated dosing. They responded: "This study focused on basic technical advantage of DREADD/CNO such as temporal and repetitive silencing." This is exactly right, and their method is best suited to provide insight into the core question of potential receptor availability, and it is essential information to evaluate the technical advantage of DREADD/CNO in repetitive silencing. They should perform PET imaging after successive days of CNO administration. Their other comment: "Successful silencing is somewhat beyond the current status of DREADD technology in monkeys." seems in contradiction to their own result, so clarification on this comment is essential.

R: The reviewer requested that we should clarify the potential tachyphylaxis after repetitive CNO dose. We agree with the importance of this issue. Based on the PET data obtained after a sequence of behavioral experiment with repetitive doses (6~8 times) of CNO in two monkeys (#171 and #184), a high level of DREADD receptor expression was maintained at rmCD (>120% striatal baseline, e.g., Fig. 2b, day 95). This indicates that tachyphylaxis does not occur after repetitive dosing in our testing condition (3 mg/kg, ~8 times, one per week).

We mention this in the current manuscript, as follows:

"After several CNO treatments, [¹¹C]CLZ-PET still showed increased uptake in the bilateral rmCD (#171, 121%, day 95, Fig. 2b; #184, 125%, day 83 after 2nd injection), suggesting that repetitive CNO treatment in our study did not produce significant on-going tachyphylaxis of hM4Di-DREADD receptors." (Pg 8, last line).

REVIEWERS' COMMENTS:

Reviewer #2 (Remarks to the Author):

The authors have addressed all my concerns.

Reviewer #3 (Remarks to the Author):

The authors have addressed my concern but the timescale of DREADD activation once a week is insufficient to rule out tachyphylaxis on a shorter timescale, which is still certainly possible given that the receptor expression would have a week to recover if it was observed. I suggest that they qualify their statement:

"After several CNO treatments, [

11

C]CLZ-PET still showed increa

sed uptake in the bilateral

rmCD (#171, 121%, day 95, Fig. 2b; #184, 125%,

day 83 after 2nd injection), suggesting that

repetitive CNO treatment in our study did not

produce significant on-going tachyphylaxis of

hM4Di-DREADD receptors." (Pg 8, last line).

To include the disclaimer that it does not rule out tachyphylaxis on a shorter time frame.

Reviewer #3 (Remarks to the Author):

The authors have addressed my concern but the timescale of DREADD activation once a week is insufficient to rule out tachyphylaxis on a shorter timescale, which is still certainly possible given that the receptor expression would have a week to recover if it was observed. I suggest that they qualify their statement: "After several CNO treatments, [¹¹C]CLZ-PET still showed increased uptake in the bilateral rmCD (#171, 121%, day 95, Fig. 2b; #184, 125%, day 83 after 2nd injection), suggesting that repetitive CNO treatment in our study did not produce significant on-going tachyphylaxis of hM4Di-DREADD receptors." (Pg 8, last line).

To include the disclaimer that it does not rule out tachyphylaxis on a shorter time frame.

R: According to the request by Reviewer #3, we have added the disclaimer that tachyphylaxis is not ruled out on a shorter time frame, as follows.

"DREADD activation once a week is insufficient to rule out tachyphylaxis on a shorter timescale." (Pg 9, line 3)